# Tropical wetlands and land use changes: The case of oil palm in neotropical riverine floodplains

Vera Camacho-Valdez[1]*, Rocío Rodiles-Hernández[2], Darío A. Navarrete-Gutiérrez[3], Emmanuel Valencia-Barrera[3]

1 Departamento de Conservación de la Biodiversidad, CONACYT—El Colegio de la Frontera Sur, San Cristóbal de las Casas, Chiapas, México, 2 Departamento de Conservación de la Biodiversidad, El Colegio de la Frontera Sur, San Cristóbal de las Casas, Chiapas, México, 3 Departamento de Observación y Estudio de la Tierra, la Atmósfera y el Océano, El Colegio de la Frontera Sur, San Cristóbal de las Casas, Chiapas, México

* vcamacho@ecosur.mx

**Data Availability Statement:** All relevant data are within the manuscript.

**Funding:** Conectividad y Diversidad Funcional de la Cuenca del rio Usumacinta" (FID-ECOSUR-784-

## Abstract

Oil palm plantations are expanding in Latin America due to the global demand for food and biofuels, and much of this expansion has occurred at expense of important tropical ecosystems. Nevertheless, there is limited knowledge about effects on aquatic ecosystems near to oil palm-dominated landscapes. In this study, we used Landsat 7 ETM+, Landsat 8 OLI imagery and high-resolution images in Google Earth to map the current extent of oil palm plantations and determined prior land use land cover (LULC) in the Usumacinta River Basin as a case-study site. In addition, we assess the proximity of the crop with aquatic ecosystems distributed in the Usumacinta floodplains and their potential effects. Based on our findings, the most significant change was characterized by the expansion of oil palm crop areas mainly at expenses of regional rainforest and previously intervened lands (e.g. secondary vegetation and agriculture). Although aquatic ecosystem class (e.g. rivers, lagoons and channels) decreased in surface around 3% during the study period (2001–2017), the change was not due to the expansion of oil palm lands. However, we find that more than 50% of oil palm cultivations are near (between 500 and 3000 m) to aquatic ecosystems and this could have significant environmental impacts on sediment and water quality. Oil palm crops tend to spatially concentrate in the Upper Usumacinta ecoregion (Guatemala), which is recognized as an area of important fish endemism. We argue that the basic information generated in this study is essential to have better land use decision-making in a region that is relative newcomer to oil palm boom.

## Introduction

The African oil palm (*Elaeis guineensis*, family Arecaceae) is a tropical forest palm native to West and Central African forests that has expanded into forest-rich developing countries and is now present in more than 16 of them [1, 2]. Indonesia is the world's largest and most rapidly

1004). The funders had no role in study design, data collection and analysis, decision to publish, or preparation of the manuscript. The authors received no specific funding for this work.

**Competing interests:** The authors have declared that no competing interests exist.

growing producer with 75 percent of total mature palm area and 80 percent of total oil palm production [3, 4]. Its annual production increased from 168,000 tons in 1967 to 22 million tons by 2010 [5] and by the end of 2020 the production increased further to 48.3 million tons [4, 6]. The main reason for this unprecedented expansion is the increase in global demand for oil palm as a source of fats and oil for human consumption, nonedible products, and biofuel feedstock to keep pace with human population growth [7–9]. Driven by increasing demand, it is projected that the global oil palm production will reach 8.9 billion tons in 2050 [10, 11].

Oil palm plantations have generated fundamental benefits to human wellbeing, providing jobs and incomes to millions of people in developing countries [3], and improving the livelihoods of local smallholders [12–14]. However, the expansion of this crop comes with several environmental and socio-economic impacts [6]. Past research has shown that oil palm expansion can trigger deforestation [15–19], loss of biodiversity [20–23], peat swamps degradation [24–26], high greenhouse gas (GHG) emissions [27] and water pollution [28], among other effects. Moreover, the loss of biodiversity and changes in ecosystem functions can lead to a decrease in the provision of important ecosystem services for local human well-being [29, 30].

Due to increasing demand for oil palm and because available land for new oil palm plantations in Southeast Asia is shrinking, new frontiers of expansion in other regions in the world have been opened [10]. In Latin America oil palm cultivation has doubled since 2001 [4] and due to increasing global demand the region is expected to become the next oil palm expansion frontier. Today, the region contains three of the top ten producing nations in the world [i.e. Colombia, Ecuador, and Honduras; 31]

In the last decade, oil palm production in the transboundary basin of the Usumacinta shared between Mexico, Guatemala and Belize has steadily increased and generated concerns over possible negative social and environmental impacts [31, 32]. In the Mexican Usumacinta region, palm is expanding rapidly, driven by government policies (e.g. productive reconversion), international investments and the presence of large areas suitable for this crop [33], highlighting the southern Lacandon rainforest (the Benemérito de las Américas and Marqués de Comillas municipalities) as the most important oil palm zone primary managed by smallholders [34]. In the case of Guatemala, the government (through the Ministry of Agriculture and Livestock) began to formally promote smallholder oil palm cultivation in 2007 through the ProPalma program, primarily in the northern territories [35]. The program was rooted in the rationale that the crop would generate development in some of the poorest parts of Guatemala and curb the problem of land sales [36]. These zones are recognized worldwide for their rich biodiversity, as they host a multitude of aquatic ecosystems and a vast extent of tropical forest cover [37, 38], which are a vital habitat for many animal and plant species [39]. However, research on environmental threats related to oil palm plantations in this region remains extremely limited [40].

Riverine floodplains are dynamic and heterogeneous ecosystems showing temporal and spatial flood variability [41]. They may contain a complex of different wetland types, which provide an extraordinary amount of unique and important ecosystem functions and thus ecosystem services like biodiversity support, water quality improvement, flood control, and carbon storage [42, 43]. Due to their heterogeneous environmental characteristics, these ecosystems are suitable for cultivation of oil palm near them [44], which could have negative effects on their environmental integrity. In particular, the use of large amounts of agrochemicals might represent a potential risk for the integrity of aquatic ecosystems and hydrological functions, and, in turn, limit the access to daily basic needs for local communities, e.g., fishing areas, food and clean water [45, 46]. Nevertheless, there is a limited knowledge about aquatic ecosystems near oil palm-dominated landscapes [21, 47]. Therefore, there is an urgent need to address this issue, not only because of the ecological importance of aquatic ecosystems

and their riparian areas but because local communities are highly dependent on freshwater resources [28, 48].

Considering the rapid expansion of oil palm plantations in the Usumacinta River Basin (URB), it is necessary to ensure the integrity of the floodplain ecosystems present in this region and the provision of ecosystem services to local communities, which depend on local extraction of natural resources for their livelihoods. To achieve this, it is imperative to generate basic information that could help decision makers, land managers and conservation organizations to guide better management of future land use in these socially and environmentally relevant landscapes. Thus, we examine the current extent, spatial pattern and vertical distribution of oil palm crops, proximity of the crops to aquatic ecosystems and their potential effects on the ecosystem services provided by the URB, as a case-study site for the present research, which includes two regions with large extensions of oil palm crops, represents a biodiversity hotspot [39, 49] and hosts considerable archaeological and cultural wealth [50].

The study aimed to answer the following questions: 1) What are the ecosystem services provided by riverine floodplain ecosystems and the potential effects of oil palm crops? 2) What is the current extent of oil palm? 3) What land use/land cover (LULC) were there before the oil palm crops? 4) How close are aquatic ecosystem to oil palm crops? 5) Do oil palm crops differ in size and spatial distribution by freshwater ecoregion? 6) What is the vertical distribution of oil palm crops?

## Study area

The transboundary river basin of the Usumacinta extends from northwestern Guatemala to the states of Chiapas and Tabasco, in Mexico, where it drains into the Gulf of Mexico (Fig 1). It is located between 16˚04' and 18˚41' N latitude and 90˚19' and 93˚00'W longitude, and covers a total area of more than 7 million hectares, with 58% remaining in the territory of Guatemala and the rest in Mexico. It is one of the regions with the highest rainfall in Mesoamerica where the average annual precipitation can reach more than 2,500 mm and the annual mean temperature is 24 ˚C [51]. A high number of important aquatic ecosystems can be found in this region due to the large amounts of water that flow through its hydrological network [52]. These aquatic ecosystems are mainly situated in two important floodplains, a deltaic one located in the lower part, and another formed by the great tributaries of the rivers Lacantún, Pasión and Negro (Salinas) in the upper URB. Both are located on low-altitudinal gradient and influenced mainly by lateral overspill of rivers.

The diversity of biotic and abiotic factors gives rise to a biodiversity considered among the highest in the world, highlighting their large remnants of forest cover as well as multitude of lakes, lagoons and marshes, which ensure the concentration of many animal and plant species [53]. The climate in the river basin varies from temperate subhumid in the mountainous regions to warm humid in the plains [54]. The population settled in the region is around 1,776,232 inhabitants mostly located in the upper parts of the river basin and distributed in more than 7,000 localities [50].

Socially, the territory hosts the Tojolabal, Tzeltal, Chol and Maya Lacandon ethnic groups in Mexico as well as Aguacateca, Quiché, Sacapultekas, Achíes, Qeqchíes, Ixiles and Mames ethnic groups in Guatemala, with a high degree of marginalization and poverty [39]. Most of the territory of the URB has a predominantly agricultural economy.

In the last decades, Mexican and Guatemalan governments, through granting small subsidies and credit, began to incorporate smallholders into the palm-oil agro-industrial chain in the URB. Through their "Productive Reconversion" (Mexico) and "Propalma" (Guatemala) programs both countries distributed large amounts of free oil palm seedlings and fertilizers to

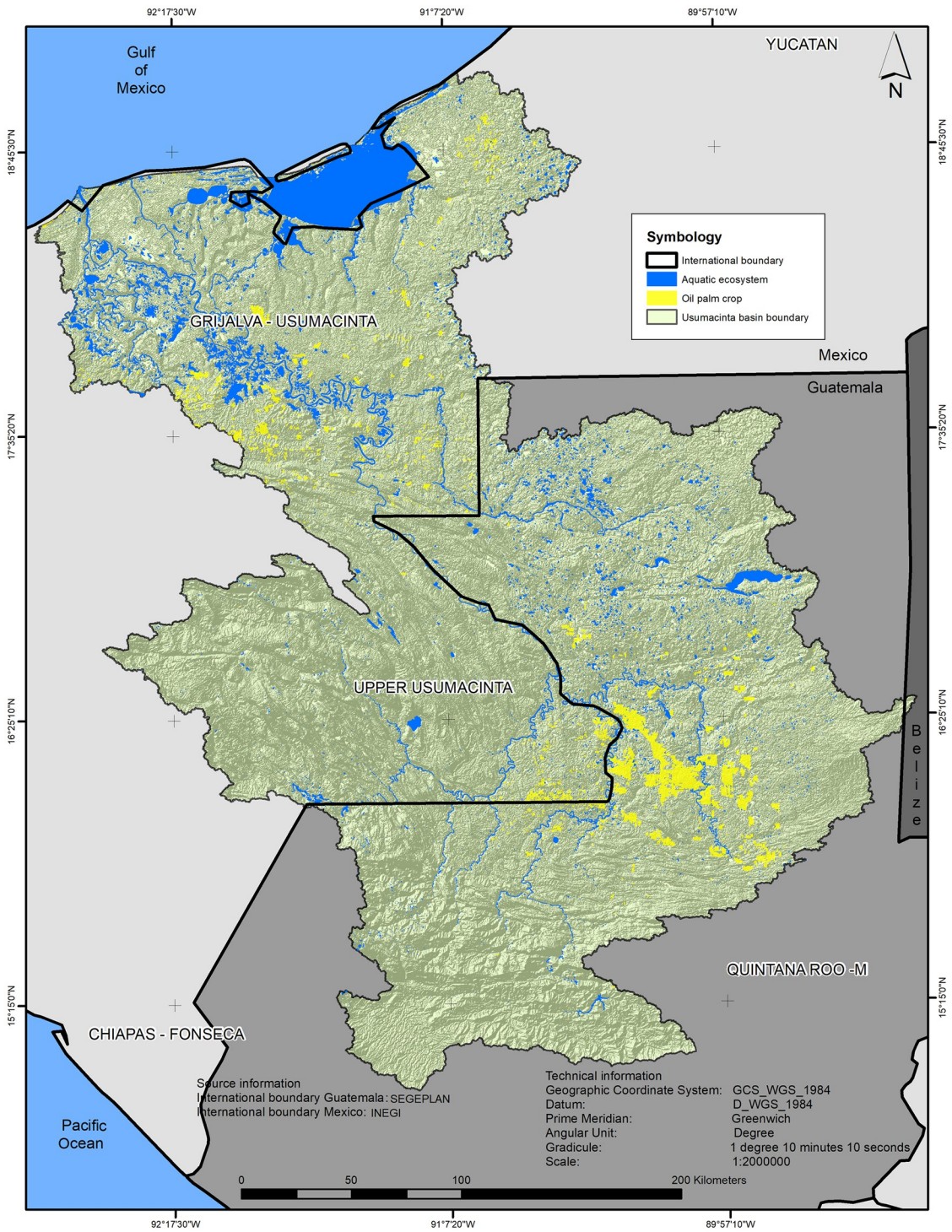

**Fig 1. Study area.** Note: For a better representation, in this map we include the polygons of the aquatic system used in the Land use/ land cover (LULC) analysis as well as the drainage network (lines). (Data source: Digital Elevation Model courtesy of the U.S. Geological Survey and drainage network for Mexico and Guatemala courtesy of INEGI and SEGEPLAN, respectively).

communities, which allowed most smallholders to enter oil palm production as independent growers [34, 55].

## Methods

The methodological approach used in this research includes the following steps: (i) the definition of riverine floodplain ecosystem services based on a global literature review, ii) analysis of land use and land cover change, and iii) spatial analysis and characterization of oil palm.

### Ecosystem services and riverine floodplains

The classification and importance of ecosystem services depends on the socioeconomic and environmental characteristics in a given region [56]. In this paper, due to lack of local information, the ecosystem services provided by riverine floodplains and the potential effects of oil palm crops on these important ecosystems were defined through a global literature review. The works performed by [16, 42, 46, 57–61], served as a basis for reviewing and defining the ecosystem services. Because it is beyond the scope of this study to give a comprehensive account of the various approaches in classifying the ecosystem services, we followed the widely accepted framework proposed by the [57], grouped into provisioning, regulating, cultural and supporting services. *Provisioning services* are goods directly enjoyed or consumed. *Regulating services* are the benefits obtained from the regulation of ecosystem processes. *Cultural services* are nonmaterial benefits that humans obtain from ecosystems. *Supporting services* are necessary for the production of all other ecosystem services.

### Land use/land cover classification

**Land use/land cover changes.** The data used to identify the distribution, estimate the size, and change of LULC present in the URB were derived from multispectral Landsat-7 ETM + and Landsat-8 OLI imagery (path /row: 19/49, 20/48; 20/49; 21/ 47; 21/ 48; 21/49, 22/47 and 22/48) which were acquired for 2001 and 2017 (https://glovis.usgs.gov/), respectively. The selection of the Landsat satellite images dates was influenced by the quality of the image especially for those with and overall cloud cover scene of 10%. Prior to interpretation, atmospheric correction was implemented to minimize contamination effects of atmospheric particles using the ATMOSC module available in IDRISI Selva software. We adopt the Cos (t) model [62] included in the ATMOSC module which incorporates all of the elements of the Dark Object Subtraction model as well as a procedure for estimating the effects of absorption by atmospheric gases and Rayleigh scattering. The corrected images used represent the proportional reflectance in real numbers with values ranging from zero to one [63]. The model is based on the cosine of the solar zenith angle (90-solar elevation). For the calibration of the images according to the type of sensor (Landsat-7 ETM+ and Landsat-8 OLI) it was based on the data of the updated calibration coefficients [64]. Each scene was geographically projected to the Universal Transverse Mercator Zone 15 North coordinate system (WGS_84 datum) and limited to the boundaries of the URB using a masking process.

The LULC classification was performed following a supervised method with the maximum likelihood algorithm. Training site data, which are areas that are known to be representative of a particular land cover type, were digitized on-line from a color composite scene. The maximum likelihood algorithm then uses the spectral signatures from these training areas to classify the whole image, which assumes that the statistics for each class in each band are normally distributed and calculates the probability that a given pixel belongs to a specific class. Each pixel is assigned to the class that has the highest probability (that is, the maximum likelihood) [63].

The validation of the output maps was assessed by an error matrix and the Kappa index (K'). All the classes were generally accurately classified (over 0.80 in both processes).

Oil palm polygons were digitized using free high-resolution imagery available from Google Earth of sufficient resolution to identify visually the pattern of individual oil palm trees. We used the confirmed oil palm polygons to determine the land use in these areas prior to oil palm expansion.

## Oil palm spatial analysis

For the spatial analysis of oil palm, the URB was divided using the Freshwater ecoregions map (https://www.feow.org/). This global map of freshwater ecoregions is based on the distributions and compositions of freshwater fish species and incorporates major ecological and evolutionary patterns [65]. The ecoregions identified were overlapped with the oil palm polygons (2017) in order to analyze their spatial distribution.

Once the thematic map (2017) was divided into freshwater ecoregions, the proximity of oil palm crops to riverine floodplains (aquatic ecosystem polygons) was calculated using the Buffer ArcGis tool. Based on other studies we include 500, 1000, 2000 and 3000 m distance [66]. We also analyzed the distribution of oil palm covers at 200 m elevation intervals.

**Landscape analysis.** To evaluate the differences between the composition and configuration of the landscape before and after the establishment of oil palm, we defined ten subsections (windows) distributed throughout the study area. To obtain the subsections (windows) we first use a grid of 2 X 2 km to cover the entire study area. We then increase the size of the grid in order to include the oil palm surface, finally we obtain for the analysis a grid of 24 X 24 km. The criteria to select the subsections (windows) were: 1) that the oil palm surface in each of the subsections was at a minimum distance of 200 m and a maximum of 500 m from aquatic ecosystems; 2) that there was an increase in oil palm cultivation and a reduction in primary vegetation based on the LULC changes obtained; 3) that the selected subsection was distributed in the Grijalva-Usumacinta and Upper Usumacinta ecoregions. The size of the subsection (windows) was determined so that they could be proportionally distributed according to the surface of each ecoregion, as well as by the extension of the oil palm inside the subsection (window). Once the subsections (windows) were defined, we use the free FRAGSTATS software version 4.2.1 [67] to quantify spatial patterns by computing landscape metrics in each subsection. For the selection of the metrics, we consider the fragmentation of the landscape and the spatial heterogeneity within each subsection and the changes over time. According to the issues of redundancy of the information provided by the landscape metrics, we used only 9 of them for the fragmentation analysis (Number of Patches–NP, Patch Density–PD, Largest Patch Index–LPI, Total Edge–TE, Edge Density–ED, Percentage of Landscape–PLAND), Landscape shape index–LSI, Interspersion-Juxtaposition Index–IJI, Effective Mesh Size–MESH. In addition, for the heterogeneity and diversity landscape analysis we used SIDI y SHE metrics [68, 69] (Table A in S1 Table).

**Statistical analysis.** Standardized Principal Component Analysis (PCA) was performed to assess changes in different areas of the Usumacinta watershed, considering the metrics calculated for both the class and landscape levels. This analysis was performed following [68]. The assumptions of sphericity, sample adequacy, and positive determinant of the matrix were previously tested. The Bartlett chi-square and the KMO tests [70] showed that the data matrices were suitable for PCA analysis. According to the Kaiser's rule (eigenvalues <1) we used the first two components for both the classes by time period and landscape level analyzes. Statistical analyzes were performed with the R software version 4.1.2 [71].

## Results

### ES identified and the potential effect of oil palm crops

**Ecosystem service provided by riverine floodplains.**   Ecosystem services defined as the benefits people obtain from the ecosystems [72], have gained attention as a tool to better understand the relationship between different influences on ecosystems and the availability of their functions as they relate to provision of services for humans [60]. Well-functioning riverine floodplains offer a broad set of provisioning, regulating, cultural and supporting services. In fact, floodplains defined as "areas of low-lying land that are subject to inundation by lateral overflow water from rivers or lakes with which they are associated" [73], contribute more than 25% of all terrestrial ecosystem services, although they cover only 1.4% of the land surface area [42, 74].

We generated a list of 18 ecosystem services provided by riverine floodplain ecosystems, which are the most representative in the reviewed global literature (Table 1). The main services include food, disturbance regulation, water supply, maintenance of water quality, nutrient cycling, recreation and tourism, among others [42, 60]. For example, complex and dynamic channel patterns in floodplains are essential for regulating flood pulses and increasing water storage [61]. Reduction in flow velocity also causes deposition of sediments, which improves water quality, supports nutrient cycling, increases productivity and improves fish habitat [60]. Moreover, riverine floodplains play a significant role in the hydrological cycle and hence the supply of water for people and the many uses they make of it, including irrigation, energy, and transport [57]. Hence, riverine floodplains are fundamental for the well-being of communities, which depend on extraction of local natural resources for their livelihoods.

**Main effects of oil palm crops on riverine floodplain ecosystems.**   Floodplains are among the world's most highly modified landscapes. The increasing intensification of land use and the associated channelization, urbanization, intensive agriculture, damming and hydropower development have led to a shift of the ecological functioning and ecosystem services provided by river landscapes [73, 74]. For example, the loss and degradation of floodplains has reduced their natural ability to buffer or ameliorate the impacts of floods [57].

The main potential impacts of oil palm crop in riverine floodplains, comes from pollution and habitat fragmentation due to deforestation [45, 46], which lead to natural habitat loss for many species and biodiversity reduction [75]. The loss or degradation of riparian vegetation through oil palm expansion has negative effects for aquatic ecosystems functioning; changes in the hydrological, biochemical and physical processes can interact and compromise stream structure and function [76] and substantially degrade the value of streams as habitats for biota [77]. In fact, loss of vegetation may modify channel cross-sectional size and shape, which may affect habitat complexity [78], connectivity [79] and stream food webs [80]. Furthermore, a reduction in riparian cover can diminish shading and promote algal growth, alter water chemistry and increase water temperature [79, 81–84]. Consequently, distribution, reproduction and trophic dynamics of aquatic species could be affected [85]. Loss of riparian vegetation also decreases the ability of the ecosystem to hold rainfall, and water is flushed more quickly into the rivers, increasing flooding in the rainy season and drought during the dry season [46].

The use of agrochemicals, such as fertilizers and pesticides, might represent a potential risk for the integrity of aquatic ecosystems and hydrological functions when agricultural practices are not optimized [16]. The major nutrients required for oil palm are mainly nitrogen, phosphorus and potassium, which could have potential effects on water quality and aquatic biota [46, 86]. For example, nitrate and phosphorous may eutrophicate aquatic ecosystems causing undesirable algal blooms, blocking sunlight and oxygen diffusion to aquatic life [47]. In

**Table 1. Ecosystem services of riverine floodplains.** Based on a literature review.

| | Services | Description | Reference |
|---|---|---|---|
| **Provisioning** | Food | Production of fish, wild animals, cultivated crops, plant resources for agricultural use | MEA (2003) |
| | | | Van der Ploeg and de Groot (2010) |
| | | | Hornung et al. (2019) |
| | Water supply | Storage and retention of water; surface and ground water for consumptive use (drinking, domestic use, agriculture, and industrial use); water for non-consumptive use (generating power, transport, and navigation) | MEA (2003) |
| | | | Van der Ploeg and de Groot (2010) |
| | | | Böck et al. (2018) |
| | | | Hornung et al. (2019) |
| | Fiber and fuel | Production of timber, fuel wood, peat, fodder, aggregates; fibers and other materials from plants for direct use or processing | MEA (2003) |
| | | | Böck et al. (2018) |
| | | | Hornung et al. (2019) |
| | Biochemical products | Extraction of materials from biota | MEA (2003) |
| | Genetic materials | Medicinal resources; genes for resistance to plant pathogens, ornamental species | MEA (2003) |
| | | | Van der Ploeg and de Groot (2010) |
| **Regulating** | Climate regulation | Retention of greenhouse gas emission/carbon sequestration; temperature regulation/cooling; precipitation regulation and other climatic processes; chemical composition of the atmosphere | MEA (2003) |
| | | | Van der Ploeg and de Groot (2010) |
| | | | Hornung et al. (2019) |
| | Maintenance of water quality | Riverine wetlands further improve water quality by reducing nitrogen, phosphorus and sulfur concentrations through plant growth, soil adsorption and anaerobic processes; natural filtration and water treatment; retention, recovery, and removal of excess nutrients and pollutants | MEA (2003) |
| | | | Böck et al. (2018) |
| | | | Hornung et al. (2019) |
| | Erosion protection | Erosion control through water/land interactions; mass flow/sediment regulation; soil formation in floodplains | MEA (2003) |
| | | | Böck et al. (2018) |
| | | | Hornung et al. (2019) |
| | Disturbance regulation | Buffering of flood flows; floodplains and associated wetlands act as a sponge and regulate water volume, releasing water during low-flow conditions; storm protection | MEA (2003) |
| | | | Van der Ploeg and de Groot (2010) |
| | | | Böck et al. (2018) |
| | | | Hornung et al. (2019) |
| **Cultural** | Spiritual and Inspirational | Personal feelings and well-being (physical and mental health benefits); religious significance; personal satisfaction from free-flowing rivers; inspiration for culture, art and design | MEA (2003) |
| | | | Van der Ploeg and de Groot (2010) |
| | | | Böck et al. (2018) |
| | | | Hornung et al. (2019) |
| | Recreation and tourism | Opportunities for tourism and recreational activities (river rafting, kayaking, hiking, and fishing) | MEA (2003) |
| | | | Böck et al. (2018) |
| | | | Hornung et al. (2019) |
| | Aesthetic | River viewing; landscape aesthetics | Böck et al. (2018) |
| | | | Hornung et al. (2019) |
| | Educational | Opportunities for formal and informal education and training | MEA (2003) |
| | Natural and cultural heritage | Historic and archaeological sites | Hornung et al. (2019) |
| **Supporting** | Biodiversity | Habitats for resident or transient species; floodplains are critical for maintaining aquatic and riparian biodiversity; most rivers are also reliant upon their floodplains to maintain fish productivity | Tockner and Stanford (2002) |
| | | | MEA (2003) |
| | Soil formation | Sediment retention and accumulation of organic matter | MEA (2003) |
| | Role in nutrient cycling and food webs | Storage, recycling, processing, and acquisition of nutrients; seasonal fluctuations in water flows distribute sediment, nutrients, seeds and aquatic organisms longitudinally through river and stream systems and laterally across active channels and floodplains; maintenance of floodplain fertility | MEA (2003) |
| | | | Van der Ploeg and de Groot (2010) |
| | | | Böck et al. (2018) |
| | Pollination | Support for pollinators | MEA (2003) |

addition, palm oil mill effluents (POME), a polluted mix of crushed shells, water, and fat residues are often released into the rivers without treatment causing a degradation of the aquatic life (e.g. fish) and drinkable water quality [45, 46, 87]. Therefore, these impacts can cause a decrease in the supply of vital ecosystem services for local communities [79].

## Analysis of land use/land cover change

Land use/land covers identified in the URB from Landsat scenes are shown in Table 2. The classification includes eleven informational classes that correspond to agricultural, oil palm crop, aquatic ecosystem, mangrove, shrubland, rainforest, dry forest, bare soil, hydrophytic, secondary vegetation and anthropogenic infrastructure. Considering the classification results, agricultural, rainforest and secondary vegetation are the dominant cover types in the study area (Table 3).

The changes in the area of each of the ten generic LULC categories, between 2001 and 2017, are represented in Table 3 and Fig 2. During 2001, the category "rainforest" comprised the largest land-cover proportion (40.65%) in the study area followed by secondary vegetation (21.64%). Other common types of LULC were agricultural, shrubland, dry forest and bare soil accounting for around 9.28%, 8.40%, 6.94%, 4.98%, respectively. Four additional LULC types (oil palm crop, aquatic ecosystem, mangrove and hydrophytic vegetation) accounted together for 6.45% of the total area.

Overall, the main LULC types that showed an increase during the period 2001–2017 were agricultural, oil palm crop, hydrophytic vegetation and secondary vegetation. The most significant change in the URB was characterized by the expansion of oil palm crop areas, which increased from 2,987 to 134,197 ha (Table 3). The results also show that bare soil decreased by 56%, while areas of wetland (aquatic ecosystems and mangrove), forest (rainforest and dry forest) and shrub land decreased by 3%, 4%, 6%, 17% and 14%, respectively.

**Table 2. Types of land use/land covers in the Usumacinta watershed.**

| ID | Class | Description |
|---|---|---|
| 1 | Agricultural | Induced land covers: agricultural, livestock and grassland |
| 2 | Oil palm crop | Oil palm trees (*Elaeis guineensis*) |
| 3 | Aquatic ecosystem | Permanent riverine wetland: river polygons, lagoons, channels |
| 4 | Mangrove | Forested-shrub estuarine wetland: plant association formed by one or a combination of the four species of mangrove |
| 5 | Shrubland | Shrub-dominated plant communities |
| 6 | Rainforest | Tropical evergreen forest: trees up to 25 m or more tall, of very diverse species and that retain their foliage all year round. It consists of vegetation such as *Dialium guianense* and *Terminalia amazonia* |
| 7 | Dry forest | Tropical deciduous forest: forests typical of regions with a warm climate and dominated by arborescent species that lose their leaves in dry seasons |
| 8 | Bare soil | Areas without vegetation: unused land, exposed soils |
| 9 | Hydrophytic vegetation | Palustrine continental wetland with more or less permanent water: swamp, marsh, tular, popal |
| 10 | Secondary vegetation | Vegetation that develops after a human or natural disturbance because of the secondary succession process. Forest characterized by a less developed canopy structure, smaller trees, and less diversity |
| 11 | Anthropogenic infrastructure | Villages, cities, roads, etc. |

**Table 3. Land use/land cover changes in the Usumacinta watershed during the period 2001–2017 (hectares).**

| Land cover type | 2001 | | 2017 | | Change (2001–2017) | |
|---|---|---|---|---|---|---|
| | Area (ha) | % | Area (ha) | % | Area (ha) | % |
| Agricultural | 834,952.67 | 9.28 | 1,295,581.74 | 14.41 | 460,629.07 | 55.17 |
| Oil palm crop | 2,987.10 | 0.03 | 86,351.85 | 0.96 | 83,364.75 | 2790.82 |
| Aquatic ecosystem | 359,749.38 | 4.00 | 350,018.50 | 3.89 | -9,730.88 | -2.70 |
| Mangrove | 78,793.91 | 0.88 | 75,729.42 | 0.84 | -3,064.49 | -3.89 |
| Shrubland | 755,298.96 | 8.40 | 649,355.54 | 7.22 | -105,943.42 | -14.03 |
| Rainforest | 3,655,717.37 | 40.65 | 3,420,161.30 | 38.03 | -235,556.07 | -6.44 |
| Dry forest | 623,698.16 | 6.94 | 494,801.15 | 5.50 | -128,897.01 | -20.67 |
| Bare soil | 447,747.01 | 4.98 | 252,550.91 | 2.81 | -195,196.10 | -43.60 |
| Hydrophytic vegetation | 138,462.56 | 1.54 | 183,049.11 | 2.04 | 44,586.55 | 32.20 |
| Secondary vegetation | 1,947,739.62 | 21.66 | 2,099,841.52 | 23.35 | 152,101.90 | 7.81 |
| Anthropogenic infrastructure | 147,725.19 | 1.64 | 85,430.88 | 0.95 | -62,294.31 | -42.17 |
| Total | 8,992,871.90 | 100 | 8,992,871.90 | 100 | | |

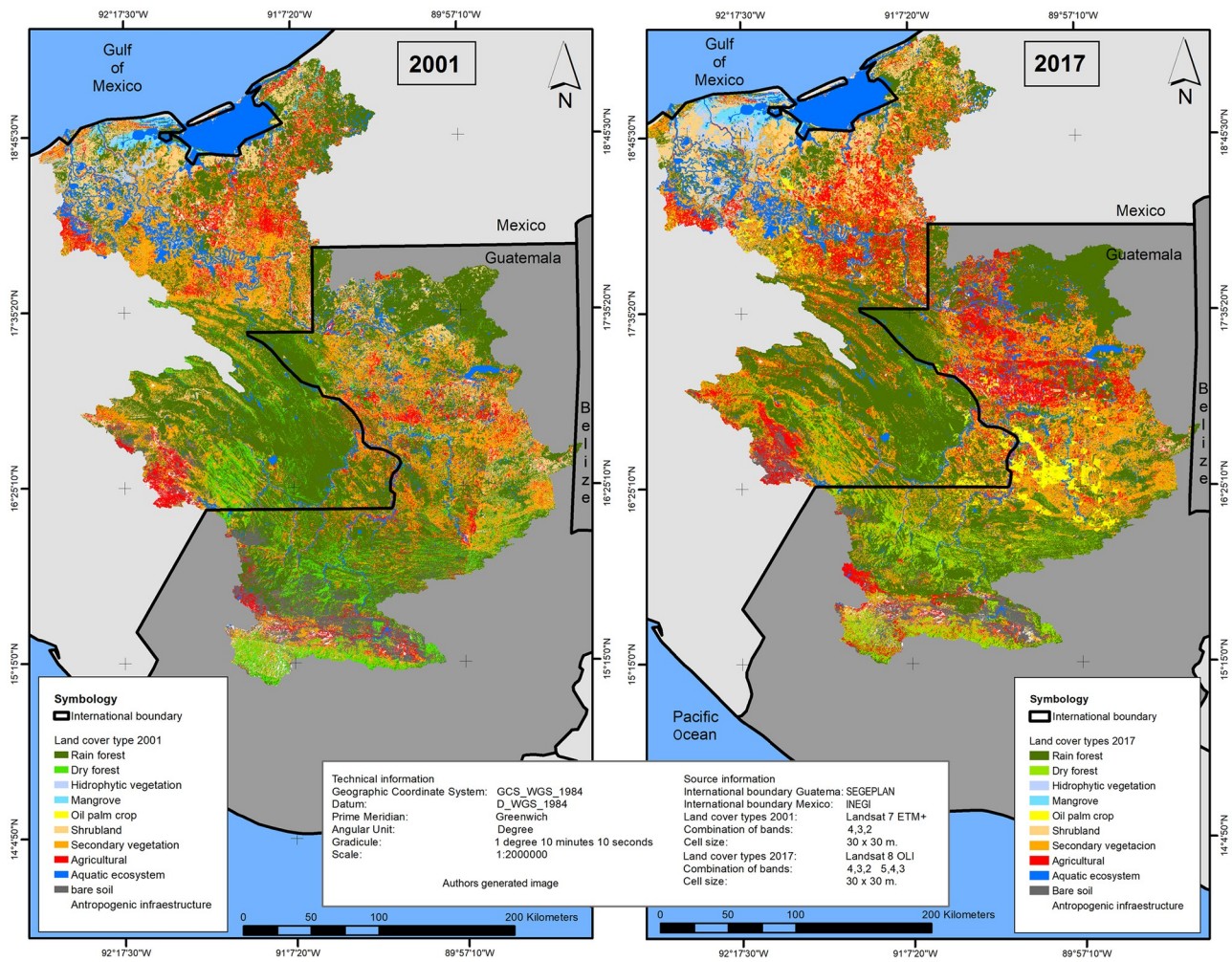

**Fig 2. Land use/land cover changes (2001–2017) generated with the GIS software ESRI ArcGIS 10.5. Copyright © 1995–2022 Esri.** All rights reserved. Published in the United States of America. (Data source: Landsat 7 ETM+ and Landsat-8 OLI image courtesy of the U.S. Geological Survey).

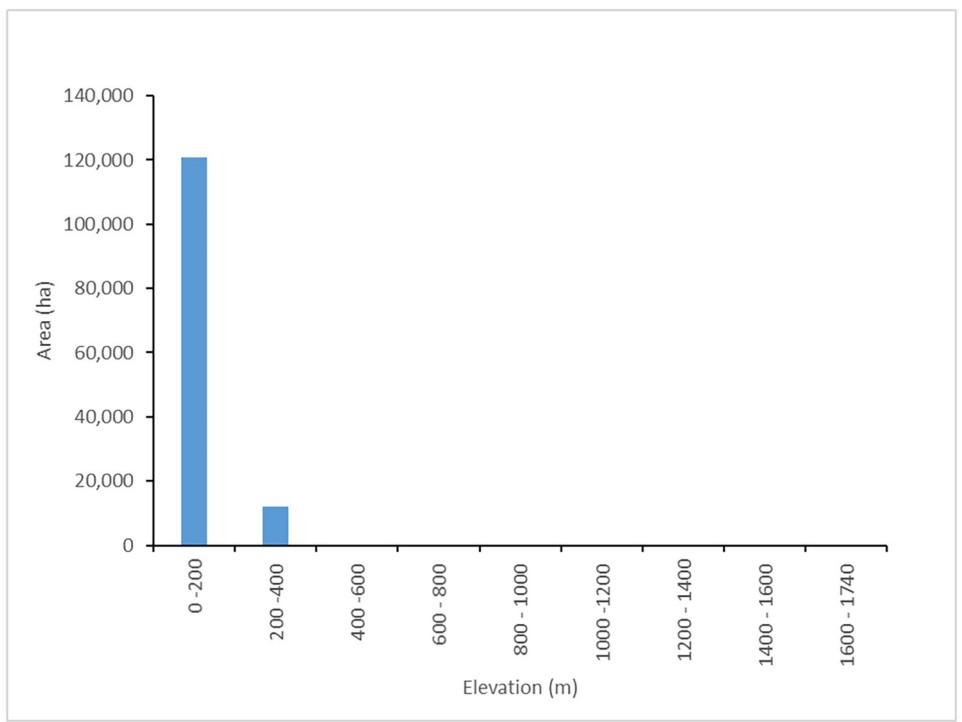

**Fig 3. Vertical distribution of oil palm crops.**

## Spatial analysis and characterization of oil palm crops

A clear altitudinal pattern was found when analyzing the proportion of Oil palm crop at 200 m elevation intervals. The total of the plantations was concentrated at elevations lower than 400 m (Fig 3).

The spatial distribution of the Oil palm crops by ecoregion exhibited important variation (Fig 4). Oil palm cultivated land expanded mainly in the Upper Usumacinta fluvial ecoregion (Guatemala-Mexico) with 107,600 ha, representing 80% of the total area. The Grijalva-Usumacinta ecoregion (Mexico) shows a lower extension of Oil palm areas with 25,000 ha, which represents 20% of the total area.

Land use/land covers that were converted to oil palm crops, are shown in Fig 5. A large proportion of rainforest areas were changed into oil palm (52,133 ha) during the study period (2001–2017), which represents 39% of the total area of oil palm (2017). Additionally, the secondary vegetation and agricultural areas converted to oil palm were substantial with 47,755 ha and 20,666 ha, respectively.

The areas of Oil palm crops near the riverine floodplains (aquatic ecosystems) are shown in Fig 6. More than 50% of Oil palm cultivated areas are located at a distance of between 500 and 3000 meters from aquatic ecosystems. Oil palm total areas are larger with a buffer of 2000 m distance (Fig 6).

**Landscape analysis.** We obtained four subsections (windows W1, W2, W3 and W4) for the Grijalva–Usumacinta freshwater ecoregion, while for Upper–Usumacinta freshwater ecoregion were six subsections (windows W5, W6, W7, W8, W9 and W10). The total area of each subsection (window) was of 57,600 ha (Figs 1–10 in S1 File).

*Land use/land cover class-level fragmentation analysis.* The landscape fragmentation analysis shows gradual changes during the study period (2001–2017). The landscape metrics-based

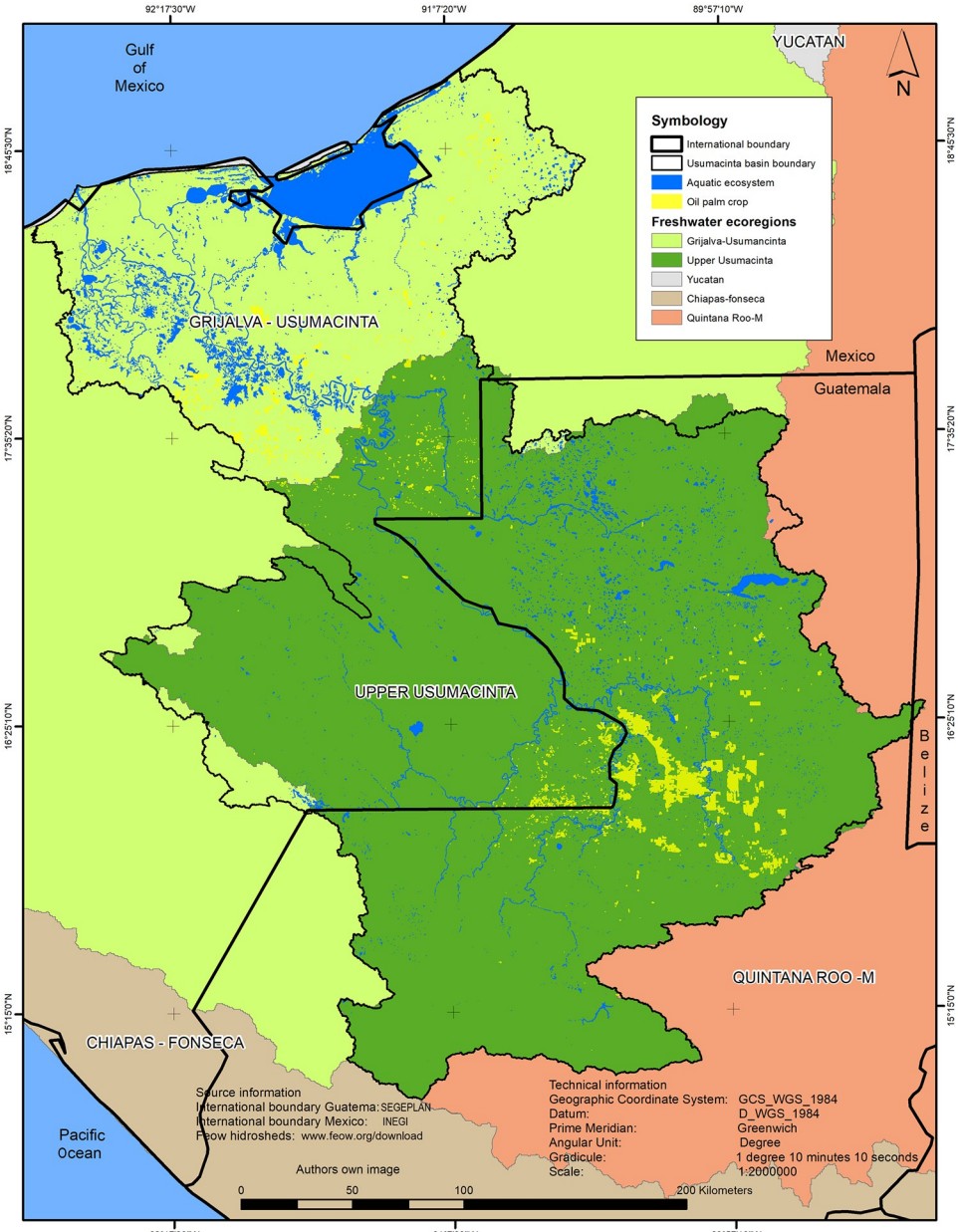

**Fig 4. Spatial distribution of oil palm crops by freshwater ecoregion (2017).** Note: For a better representation, in this map we include the polygons of the aquatic system used in the LULC analysis as well as the drainage network (lines). (Data source: Shape file of the freshwater ecoregions of the world courtesy of Freshwater ecoregions of the world (FEOW).

analysis of the two individual years by LULC classes have provided information related with how the patterns of land covers changes over time. The number of patches of rain forest, dry forest, hydrophytic vegetation and aquatic ecosystems decreased for most of the subsections (windows), with the exception of the subsection W3 in which they increased (Table B in S1 Table). In the case of land uses such as secondary vegetation, agricultural areas and bare soil, the patches increased in 2017. Whereas oil palm cover increased significantly between 2001 and 2017 in most subsections, highlighting the subsection W9 from 304 to 30,028 ha. A

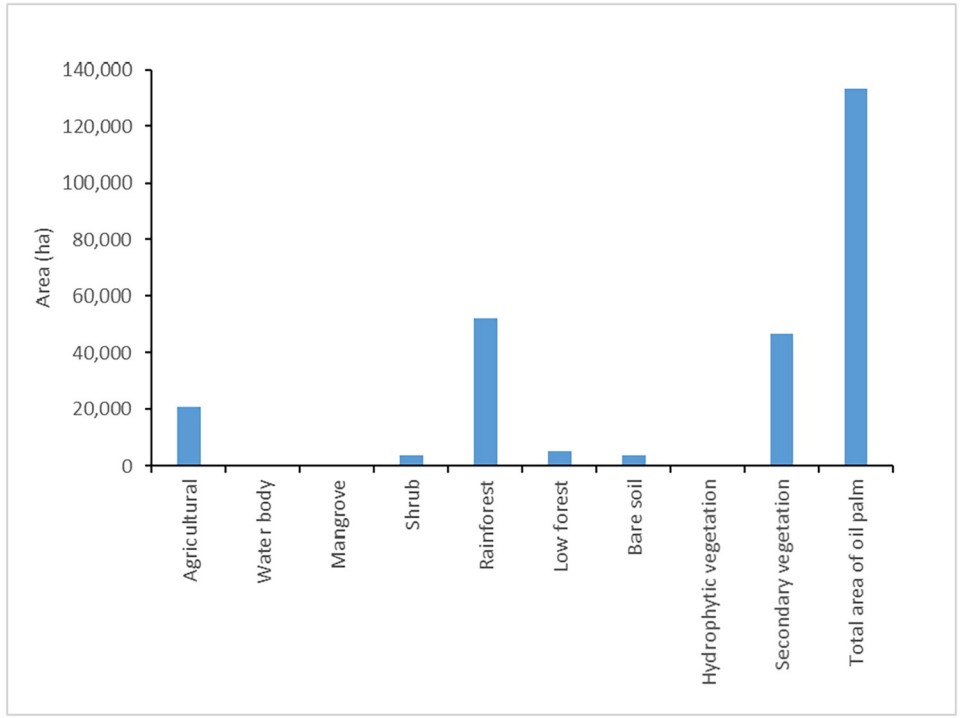

**Fig 5. LULC areas converted to oil palm crops during the study period (2001–2017).**

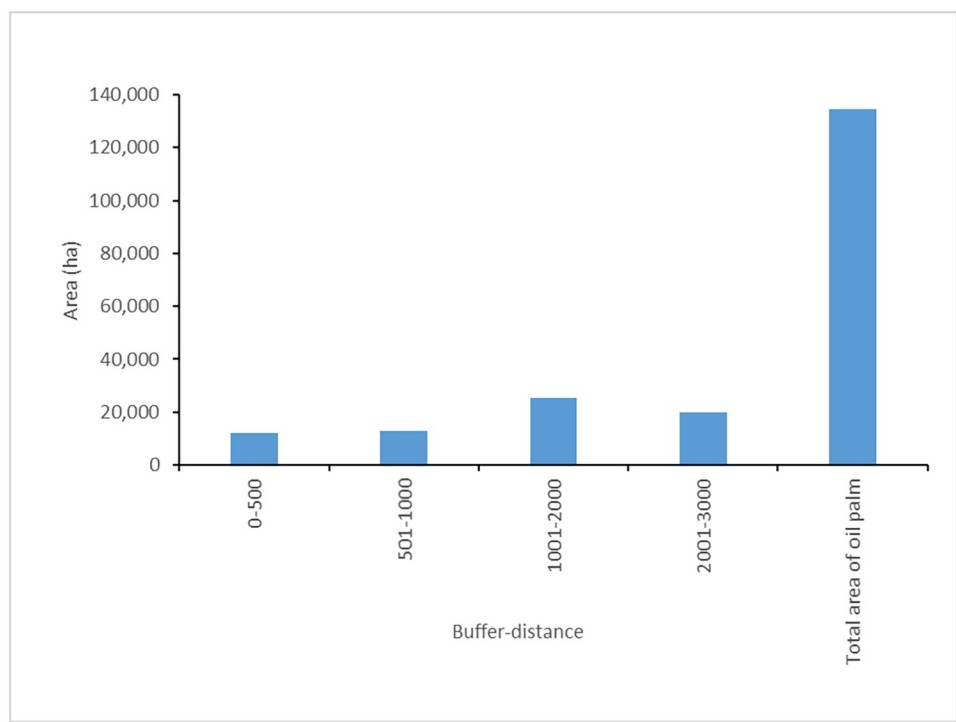

**Fig 6. Total Oil palm areas (in hectares) estimated at a distance of 500, 1000, 2000 and 3000 m.**

**Table 4. Landscape-level metrics for the ten subsections (W1-W10) during 2001–2017.**

| Window | Year | TA | NP | PD | LPI | TE | ED | LSI | CONTAG | IJI | MESH | SHDI | SIDI |
|--------|------|------|------|-------|-------|---------|-------|-------|--------|-------|----------|------|------|
| W1 | 2001 | 57600 | 6964 | **12.09** | 37.42 | 4596510 | 79.80 | 48.88 | 49.80 | 71.35 | 8230.11 | 1.46 | 0.70 |
|    | 2017 | 57600 | 6575 | **11.41** | 12.43 | 4616700 | 80.15 | 49.09 | 51.73 | 67.14 | 1298.13 | 1.59 | 0.75 |
| W2 | 2001 | 57600 | 5770 | **10.02** | 18.31 | 4130130 | 71.70 | 44.02 | 51.93 | 68.66 | 3073.80 | 1.53 | 0.72 |
|    | 2017 | 57600 | 4818 | **8.36** | 14.76 | 3497310 | 60.72 | 37.43 | 49.85 | 69.15 | 2309.08 | 1.67 | 0.77 |
| W3 | 2001 | 57600 | 6808 | 11.82 | 72.49 | 3282660 | 56.99 | 35.19 | 67.48 | 69.47 | 30364.16 | **0.99** | 0.43 |
|    | 2017 | 57600 | 9289 | 16.13 | 41.82 | 4871430 | 84.57 | 51.74 | 50.31 | 71.48 | 10240.38 | **1.54** | 0.69 |
| W4 | 2001 | 57600 | 7259 | **12.60** | 21.64 | 4368750 | 75.85 | 46.51 | 52.46 | 67.17 | 4293.74 | **1.49** | 0.71 |
|    | 2017 | 57600 | 6673 | **11.59** | 13.86 | 4016700 | 69.73 | 42.84 | 46.70 | 81.21 | 2030.41 | **1.60** | 0.75 |
| W5 | 2001 | 57600 | 5929 | **10.29** | 38.19 | 4220700 | 73.28 | 44.97 | 56.97 | 58.23 | 8615.55 | 1.33 | 0.67 |
|    | 2017 | 57600 | 4657 | **8.09** | 8.41 | 3779280 | 65.61 | 40.37 | 55.97 | 56.42 | 1143.90 | 1.40 | 0.72 |
| W6 | 2001 | 57600 | 11118 | **19.30** | 9.66 | 5497350 | 95.44 | 58.26 | 53.00 | 69.91 | 1486.53 | **1.47** | 0.69 |
|    | 2017 | 57600 | 4096 | **7.11** | 13.32 | 3256080 | 56.53 | 34.92 | 52.22 | 70.76 | 2270.97 | **1.60** | 0.75 |
| W7 | 2001 | 57600 | 8854 | **15.37** | 18.41 | 5056770 | 87.79 | 53.67 | 58.56 | 55.91 | 4298.25 | 1.21 | 0.64 |
|    | 2017 | 57600 | 6394 | **11.10** | 17.56 | 4145730 | 71.97 | 44.18 | 56.89 | 63.94 | 2564.17 | 1.33 | 0.66 |
| W8 | 2001 | 57600 | 8294 | **14.40** | 24.33 | 4471950 | 77.64 | 47.58 | 58.62 | 61.64 | 6021.28 | **1.24** | 0.64 |
|    | 2017 | 57600 | 4437 | **7.70** | 20.27 | 3261540 | 56.62 | 34.97 | 55.10 | 67.16 | 3949.85 | **1.48** | 0.71 |
| W9 | 2001 | 57600 | 8232 | **14.29** | 24.31 | 4642110 | 80.59 | **49.36** | 58.45 | 61.29 | 5371.25 | 1.23 | 0.64 |
|    | 2017 | 57600 | 2808 | **4.88** | 48.35 | 2124780 | 36.89 | **23.13** | 64.76 | 64.57 | 14138.86 | 1.21 | 0.62 |
| W10 | 2001 | 57600 | 11757 | **20.41** | 28.70 | 5743020 | 99.71 | 60.82 | 55.92 | 61.17 | 5353.78 | 1.34 | 0.67 |
|     | 2017 | 57600 | 7518 | **13.05** | 31.10 | 4240470 | 73.62 | 45.17 | 54.60 | 67.47 | 5976.58 | 1.42 | 0.68 |

dramatic change can be observed in the case of the rain forest: MESH has decreased significantly in W1, W7, W8 and W9 subsections (for more details for each LULC see Table B in S1 Table).

*Landscape-level fragmentation analysis.* The density of patches (PD), a metric that is associated with the landscape, decreased from 2001 to 2017 in all subsections (windows), which represents the homogenization of the landscape and the loss of natural complexity towards anthropic activities, mainly due to oil palm cultivation. Whereas the LPI metric increased in four subsections between 2001 and 2017 (W6, W8, W9 and W10); this increase indicates that the composition in the landscape is given by a single patch. However, for other subsections the value of this metric between the years analyzed decreases (W1, W2, W3, W4, W5 and W7). Metrics such as TE and ED showed an increase in W1 and W3. For the other subsections, the values of these two metrics decreased, which represents that the landscape has a greater number of divisions. The LSI metric did not show significant differences between both periods except for subsection W9 (Table 4).

*Multivariate analysis of changes.* The landscape-level PCA model explained 84% of the total variance. PC1 accounted for 47.7% of the variance and was in strong correlation with NP, PD, LSI and ED and heterogeneity and diversity values (SIDI and SHDI); PC2 accounted for 36.6% of the variance and was correlated with MESH, CONTAG and LPI (Fig 7, Table 4). PC1 showed for most of the subsection's changes in the structure (a smaller number of patches), less diversity, and a greater homogenization of the landscape from 2001 to 2017. Some subsections, such as W3, showed an increase in the diversity and complexity of their structure, due to the contribution of oil palm (Fig 7, Table 4). Overall, landscape configuration metric (IJI), have increased from 2001 to 2017 in most subsections. The values in PC1 are very low, indicating that in subsections W2, W4, W5 and W8 the diversity and heterogeneity of the landscape

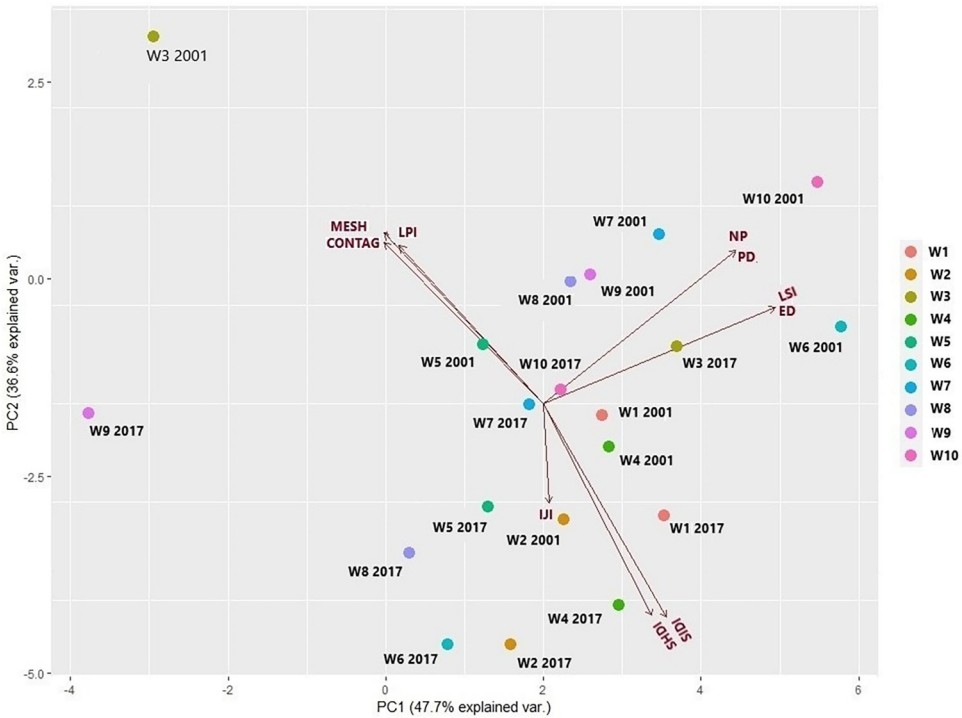

**Fig 7. Ordination plot of landscape-level metrics (colors: subsections-windows.**

decreases and patches with more homogeneous shapes increase (Table 4 and Table B in S1 Table; Fig 7).

## Discussion and conclusions

### Land use/land cover pattern

Measuring land cover changes through the combination of high-resolution imagery from Google Earth and Landsat offers relevant information for understanding the effects of agricultural development, in this case, from oil palm plantations. Based on our findings, during the last decades, the most significant LULC change in the URB was caused by the expansion of oil palm crop areas mainly at expenses of regional rainforest (Fig 4). In fact, fragmentation analyses exhibit also that changes mainly occurred in rainforest and dry forest present in the Upper Usumacinta. A decrease of important metrics in both covers represents the homogenization of the landscape and the loss of natural complexity mainly due to oil palm cultivation. This trend of deforestation and its fragmentation is common in several tropical countries where oil palm development led to widespread clearing of forested land [19, 31, 88, 89] causing significant environmental impacts. The conversion of forests to palm plantations reduces plant diversity and eliminates the animal species that depend on natural forest as well as decreasing the supply of important ecosystem services to local communities [19]. If the current expansion rate of oil palm continues, the URB which is considered an important biodiversity hotspot region, will likely have important environmental and social impacts. In fact, studies have identified that clearing tropical forests for oil palm cultivation results in strong local and regional biodiversity declines [16, 20–23]. Overall, monocultures as Oil palm are far less structurally complex than tropical forest they replace [19]; these plantations support far fewer species because they lack

the complex and rich vegetation that is needed to support the high biodiversity of tropical forests [90]. Furthermore, forest-dependent communities will be affected in their livelihoods.

Regarding the spatial analysis of aquatic ecosystems distributed in the Usumacinta riverine floodplains, we found that significant portions of oil palm areas are close to aquatic ecosystems (more than 50% of the total area are between 500 and 3000 m distance) in low-altitudinal areas of the URB. This proximity to water bodies could have significant environmental impacts. The effects on sediment and water quality can extend over comparatively large distances [91]. Increased sediment on aquatic ecosystems reduces water clarity that negatively affects aquatic plants, which are often the key primary producers in these ecosystems [92]. In addition, the reduced water clarity, substantially interfere with biological connectivity because turbid zones can act as barriers to movement of aquatic fauna [77]. Sediment inputs can therefore have substantial impacts on the structure and functioning of aquatic ecosystems, which may ultimately affect their provision of ecosystem services to local communities [93].

On the other hand, the agrochemicals used for oil palm cultivation could affect water quality, altering important environmental conditions (e.g. biochemical cycles) with adverse consequences for local people whose livelihoods depend on riverine resources [77]. The redistribution of water flows may cause periodic water scarcity in villages surrounding oil palm estates [94] and high sedimentation in disturbed aquatic ecosystems could contribute to increased downstream flood risk [46]. Furthermore, our analysis shows that oil palm lands tend to spatially concentrate in the Upper Usumacinta ecoregion (mainly northern Guatemala and southern Mexico). This region has been recognized as an area of important fish endemism due to its geographical complexity that potentially led to the isolation of populations with unique biodiversity, assemblages and molecular diversity [95]. In fact, due to its great biodiversity, a significant number of Natural Protected Areas (NPAs) are present in the region. Despite this, northern Guatemala is a vast frontier that has undergone considerable oil palm expansion in the last decade, where three companies control 41% of the total area under oil palm and smallholders play a lesser role [96]. This expansion has been associated with environmental degradation, where large portions of old-growth forests were replaced by oil palm plantations [97]. Freshwater fish habitats are known to be positively influenced by different functional mechanism from forests and so the conversion of native rainforest to oil palm plantations will likely have impacts on the in-stream fish communities in this region [98]. Therefore, in the context of oil palm expansion, mainly in the Mexican Upper Usumacinta region where oil palm expansion is less than in Guatemala, deforestation must be avoided and thus the socio-ecological costs, mention previously, will be reduced. In addition, effective aquatic conservation strategies are urgent, including the creation of narrow buffer zones between wetlands and intensive land-uses and the implementation of monitoring programs in order to maintain or improve the ecological functioning and biodiversity of aquatic ecosystems and the associated benefits for local communities [77].

## Oil palm regional development and socio-economic effects

Although Mexico is the largest importer of oil palm in Latin America, accounting for about half (4.54MMT) of total imports in the region [31]. Our results show an important expansion of the crop during the study period (2001–2017). This pattern of expansion has occurred through the strong intervention of the state, both in the Guatemalan and Mexican portion of the URB [38, 99]. In fact, the promotion of oil palm in Mexico and Guatemala was articulated as part of two state programs known as "Productive reconversion" (Mexico) and "Propalma" (Guatemala) with funding of international organizations such as the World Bank and the

Inter-American Development Bank (IDB), in order to take advantage of the oils from its fruits and seeds [36]. Arguing that oil palm can generate higher incomes for plantation workers´ households and reduce dependence on imported fossil fuels as well as contribute to climate change mitigation through the production of bioenergy [37]. Although oil palm biomass can be transformed into biofuels (e.g. bioethanol) which is considered as the main source for domestic renewable energy [100], the deforestation due to oil-palm expansion results in a significant loss of net biomass carbon, which contributes to carbon emissions [38]. For example, total carbon losses from biomass due to the conversion of tropical virgin peat swamp forest into oil palm plantations are estimated to be around 427.2 ± 90.7 t C ha−1 [101]. In fact, it would take between 75 and 93 years for the carbon emissions saved through use of biofuel to compensate for the carbon lost through forest conversion [102].

In addition, the deforestation and agrochemicals used for oil palm cultivation could cause a loss of organic matter and compaction in the soil which reduces water infiltration as well as an increase in carbon emissions [77].

Although several studies suggest that oil palm cultivation can contribute to rural development by providing economic resources to local populations [14, 103], other works have identified negative social impacts, with serious implications for rural communities food security, land concentration, loss of income and access to natural resources [9, 36, 104]. In the Latin-American context, because smallholders do not have the economic resources to access the technology required to produce high quality palm fruit, oil palm production is almost exclusively dominated by private corporations, which can contribute to rising inequality [105]. In fact, in some cases, poor farm households without sufficient access to capital and through coercive mechanisms have been forced to sell their land to these agro-business companies. In addition to grabbing land through lease or purchase, corporate plantations also expand their control over land and labor through hiring farmers without guaranteeing any labor rights, to the detriment of workers' physical and mental well-being [6, 106]. Furthermore, it has been documented that in Guatemala, households working in oil palm plantations, and particularly women, have no time for community activities, personal care, or resting [99].

The previous arguments show that oil palm expansion has originated important environmental impacts, economic incomes but not for all as well as conflicts over land. However, global demand for oil palm and other monocultures like soybean is expected to keep soaring [106]. To mitigate the environmental and social impact it is of uttermost importance to generate basic information, as the one in our study in the context of Latin America that can be employed by policy-makers. Prioritizing land conservation and an equitable distribution of economic benefits are key to establishing a more sustainable palm sector in a region that is relative newcomer to oil palm boom. Furthermore, an important policy area is to establish a clear delineation of protected forest lands and aquatic ecosystems. Ecosystems that are particularly environmentally vulnerable, such as riparian forests, certainly deserve special protection.

## Supporting information

**S1 Table.**
(DOCX)

**S1 File.**
(DOCX)

## Acknowledgments

We thank to Héctor Trejo (LAIGE-Ecosur) for his invaluable support with the spatial analysis. Authors are also grateful to Matteo Cazzanelli for English language reviewing and useful comments on the manuscript.

## Author Contributions

**Conceptualization:** Vera Camacho-Valdez, Rocío Rodiles-Hernández, Emmanuel Valencia-Barrera.

**Data curation:** Darío A. Navarrete-Gutiérrez.

**Formal analysis:** Vera Camacho-Valdez, Darío A. Navarrete-Gutiérrez.

**Investigation:** Vera Camacho-Valdez, Darío A. Navarrete-Gutiérrez.

**Methodology:** Vera Camacho-Valdez.

**Resources:** Rocío Rodiles-Hernández.

**Software:** Darío A. Navarrete-Gutiérrez, Emmanuel Valencia-Barrera.

**Supervision:** Vera Camacho-Valdez.

**Validation:** Vera Camacho-Valdez.

**Visualization:** Darío A. Navarrete-Gutiérrez, Emmanuel Valencia-Barrera.

**Writing – original draft:** Vera Camacho-Valdez.

**Writing – review & editing:** Vera Camacho-Valdez, Rocío Rodiles-Hernández, Darío A. Navarrete-Gutiérrez, Emmanuel Valencia-Barrera.

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
