## [Decision Letter · Decision Letter 0]

29 Oct 2021

PONE-D-21-26813Tropical wetlands and land use changes: The case of oil palm in neotropical riverine floodplains.PLOS ONE

Dear Dr. Camacho-Valdez,

Thank you for submitting your manuscript to PLOS ONE. After careful consideration, we feel that it has merit but does not fully meet PLOS ONE’s publication criteria as it currently stands. Therefore, we invite you to submit a revised version of the manuscript that addresses the points raised during the review process.

We look forward to receiving your revised manuscript.

Kind regards,

Bijeesh Kozhikkodan Veettil

Academic Editor

PLOS ONE

Journal Requirements:

2. In your Methods section, please provide additional location information about your study area, including geographic coordinates for the data set if available.

4. We note that Figures 1,2 and 4 in your submission contain [map/satellite] images which may be copyrighted. All PLOS content is published under the Creative Commons Attribution License (CC BY 4.0), which means that the manuscript, images, and Supporting Information files will be freely available online, and any third party is permitted to access, download, copy, distribute, and use these materials in any way, even commercially, with proper attribution. For these reasons, we cannot publish previously copyrighted maps or satellite images created using proprietary data, such as Google software (Google Maps, Street View, and Earth). For more information, see our copyright guidelines: http://journals.plos.org/plosone/s/licenses-and-copyright.

a. You may seek permission from the original copyright holder of Figures 1,2 and 4 to publish the content specifically under the CC BY 4.0 license.  

Reviewers' comments:

Reviewer's Responses to Questions

**Comments to the Author**

1. Is the manuscript technically sound, and do the data support the conclusions?

Reviewer #1: Yes

Reviewer #2: Partly

2. Has the statistical analysis been performed appropriately and rigorously? 

Reviewer #1: Yes

Reviewer #2: No

3. Have the authors made all data underlying the findings in their manuscript fully available?

Reviewer #1: Yes

Reviewer #2: Yes

4. Is the manuscript presented in an intelligible fashion and written in standard English?

Reviewer #1: Yes

Reviewer #2: Yes

5. Review Comments to the Author

Reviewer #1: This article presents valuable new information on the growing extent of oil palm plantations in a region of Latin America that has high ecological value. Such information will assist land managers and decision-makers in assessing the impacts of oil palm plantations, and to balance the trade-offs between economic development and ecosystem conservation. This article is timely given that the majority of the research focused on the impacts of palm oil plantations has been conducted in Asia, and yet as you correctly point out, such plantations have become established in other parts of the world and such trends are predicted to continue. The data reported are illustrated by high quality maps. I have made some suggestions for improvements in my detailed comments (below).

Specific comments:

Lines 15-16: To improve the language, I recommend amending "due to the global food demand and biofuels" to "due to the global demand for food and biofuels".

Line 16: The word "expenses" should be amended to "expense".

Line 17: The word "a" can be deleted.

Line 18: Amend "nearby" to "near to".

Line 20: Add the word "the" before "current".

Line 21: Add the word "a" before the phrase "case-study".

Line 25: What are "intervened lands" in this context? Those previously impacted by people?

Line 26: It is not clear what the phrase "aquatic ecosystem cover" means in this context. Do you mean the spatial extent of aquatic ecosystems? It is clearly an important concept for your study, and so it would be helpful to the reader to define it.

Lines 26-27: "Although aquatic ecosystem cover decreased in surface during the study period". The abstract would be more informative if you would state by how large this decrease was. What % of the initial area was lost? The information from Table 3 could be used here.

Line 43: Does "production" here refer to annual production? Please clarify.

Line 43: Amend "increase" to "increased".

Line 50: Amend "has" to "have".

Line 134: I recommend that here you clarify that LULC refers to "land use land cover", as readers that look at the figures first may have missed this definition in the text.

Line 344: Amend "decrease" to "decreasing".

Line 350: Change "much" to "far".

Line 351: The word "a" should be changed to "the", and the word "are" should be amended to "is".

Lines 362-363: "The effects on sediment and water quality can extend over comparatively large distances". It would be useful to explain more detail what effects increased sediment inputs might have on water quality and aquatic ecosystems. In particular, the point should be made that high sediment inputs to aquatic ecosystems can increase turbidity, which negatively affects aquatic plants, which are often the key primary producers in aquatic ecosystems (O’Hare, M.T., et al. 2018. Plants in aquatic ecosystems: current trends and future directions. Hydrobiologia, 812, 1-11.). Sediment inputs can therefore have substantial impacts on the structure and functioning of aquatic ecosystems, which may ultimately affect their provision of ecosystem services to local communities (Mitsch, W.J., et al. 2015. Ecosystem services of wetlands. International Journal of Biodiversity Science, Ecosystem Services & Management, 11, 1-4.).

Line 378: This is an important point, because fish communities are known to be influenced by the forests within their catchment (Lo, M., et al. 2020. The influence of forests on freshwater fish in the tropics: A systematic review. BioScience, 70, 404-414.), and so the conversion of native rainforest to oil palm plantations will likely have impacts on the in-stream fish communities in this region.

Lines 762-766: There is an error with the formatting of the references here. There are currently two reference #37s and two reference #38s.

Reviewer #2: Major comments:

The expansion of oil palm plantations is a global issue. This study used Landsat 8 OLI images coupled with regular geostatistics to quantify the spatial distributions of oil palm area and the land-use changes of different landscapes (agricultural, oil palm crop, aquatic ecosystem, etc..) over 17-year plantation history (from 2001-2017). As a case study, the present work alerts the audiences to the serious problem in a biodiversity hotspot (Usumacinta River Basin), in face to the accelerated oil palm plantation processes. Several major comments are raised for your consideration:

(1) Although the present study has used several geostatistical tools to reveal the landscape patterns in the Usumacinta watershed, the effects of oil palm crops on floodplain ecosystems remained largely descriptive, without substantial data supporting them (e.g., lines of the evidence mostly derived from previous pieces of literature). Therefore, the objectives of this study should be better framed.

(2) To well address the above issue, I suggest further quantifying the fragmentation degree of landscape and/or landscape heterogeneity characteristics, as well as other commonly-used geographical indices before and after large-scale oil palm plantation, based on the currently obtained data points. For instance, how “environmental integrity” is impaired by oil palm plantation, since the authors stated in multiple places the plantation could affect habitat complexity, environmental integrity, and connectivity, and highlighted the importance of structurally complex ecosystems of naturally-occurring tropical forest. By doing this, the authors can provide robust evidence to support the arguments about the negative impacts of oil palm plantations.

(3) Some technical issues should be addressed before publication: a) lines 180-181, the cloud cover data for the images used should be provided; b) Landsat 8 satellite was only launched after 2013, how can obtain images using Landsat 8 for those in 2011-2012; c) line 182, more information about atmospheric and geometrical correction should be provided. For instance, what models have been adopted for atmospheric correction?

(4) There is a combined result and discussion in the Result section. Please make a clear separation between your results and discussion, and be sure the discussion answers what the introduction asked.

Minor comments:

1. Line 44, oil palm production in 2010 and 2011 has been reported. What about the current conditions (e.g., in 2020 or the recent five years)?

2. Lines 48-49, “8.9 billion tons in 2050”, it means globally or in Indonesia? Please specify.

3. Line Abstract & 21 & 110, add “a” before case-study site

4. Lines 114-118, the study aim should be framed more logically. For example, the vertical distribution of oil palm plantations, land-use conversion, as well as the spatial distribution of oil palm crops by freshwater ecoregion have been studied, but not been clearly stated here.

5. Line 128, remove the comma before [52].

6. Line 129, add a colon between “floodplains” and “a deltaic one”.

7. Lines 169-170, rephrase this sentence.

8. Line 203, “Table 2” should not be ahead of “Table 1”.

9. Lines 251-252, your data points cannot support your statements “we found that the greatest environmental impact comes from pollution and habitat fragmentation due to deforestation”. Besides, a citation of this sentence means that you cited previous reports to support your viewpoints.

10. Line 301, from Table 3, how can the increased oil palm areas be contributed to a net loss of 39% area of rainforest?

11. Lines 354-358, it was stated that not all lost aquatic areas can be attributed to the expansion of oil palm crops. Please place more explanation about this. Was it converted into other types of land?

12. Lines 401-403, perhaps the authors can give a simple calculation to account for this conclusion. What is the balance between the production of bioenergy in form of carbon and the loss of biomass carbon?

6. PLOS authors have the option to publish the peer review history of their article (what does this mean?). If published, this will include your full peer review and any attached files.

Reviewer #1: No

Reviewer #2: No

---

## [Author Response · Author response to Decision Letter 0]

2 Mar 2022

We are pleased to resubmit a revised version of our manuscript for publication. We appreciate the time and effort put forth by the editor and the constructive criticisms of the reviewers. We have addressed the issues indicated in the review process, and we believe that the revised version meets the journal’s publication requirements. We provide responses to the comments below. 

REVIEWER 1

GENERAL COMMENTS

This article presents valuable new information on the growing extent of oil palm plantations in a region of Latin America that has high ecological value. Such information will assist land managers and decision-makers in assessing the impacts of oil palm plantations, and to balance the trade-offs between economic development and ecosystem conservation. This article is timely given that the majority of the research focused on the impacts of palm oil plantations has been conducted in Asia, and yet as you correctly point out, such plantations have become established in other parts of the world and such trends are predicted to continue. The data reported are illustrated by high quality maps. I have made some suggestions for improvements in my detailed comments (below).

SPECIFIC COMMENTS

Lines 15-16: To improve the language, I recommend amending "due to the global food demand and biofuels" to "due to the global demand for food and biofuels".

Line 16: The word "expenses" should be amended to "expense".

Line 17: The word "a" can be deleted.

Line 18: Amend "nearby" to "near to".

Line 20: Add the word "the" before "current".

Line 21: Add the word "a" before the phrase "case-study".

Response

We appreciate your review and comments. We took these suggestions into account and the changes were made in the main text accordingly.

Line 25: What are "intervened lands" in this context? Those previously impacted by people?

Response

Yes, we referred to those land use/land cover that has been disturbed naturally or unnaturally, such as grazing, tree felling, frequent fires as well as land modified for agriculture.

Line 26: It is not clear what the phrase "aquatic ecosystem cover" means in this context. Do you mean the spatial extent of aquatic ecosystems? It is clearly an important concept for your study, and so it would be helpful to the reader to define it.

Response

Yes, “aquatic ecosystem cover” refers to the spatial extent of aquatic ecosystems that includes rivers, lagoons and channels. In the abstract we change the word “cover” by “class” for a better clarity. We include also in parenthesis the ecosystems included in this class.

Lines 26-27: "Although aquatic ecosystem cover decreased in surface during the study period". The abstract would be more informative if you would state by how large this decrease was. What % of the initial area was lost? The information from Table 3 could be used here.

Response

Following the suggestions of the reviewer we have rewritten the abstract for better clarity, highlighting the percentage of change of aquatic ecosystem class. 

Line 43: Does "production" here refer to annual production? Please clarify.

Response

Yes, we referred to the annual production of palm oil in Indonesia. We include at the beginning of the paragraph the word “annual” for a better understanding of the context.

Line 43: Amend "increase" to "increased".

Line 50: Amend "has" to "have".

Line 134: I recommend that here you clarify that LULC refers to "land use land cover", as readers that look at the figures first may have missed this definition in the text.

Line 344: Amend "decrease" to "decreasing".

Line 350: Change "much" to "far".

Line 351: The word "a" should be changed to "the", and the word "are" should be amended to "is".

Response

We appreciate your review and comments. We accept these suggestions and the text was modified accordingly.

Lines 362-363: "The effects on sediment and water quality can extend over comparatively large distances". It would be useful to explain more detail what effects increased sediment inputs might have on water quality and aquatic ecosystems. In particular, the point should be made that high sediment inputs to aquatic ecosystems can increase turbidity, which negatively affects aquatic plants, which are often the key primary producers in aquatic ecosystems (O’Hare, M.T., et al. 2018. Plants in aquatic ecosystems: current trends and future directions. Hydrobiologia, 812, 1-11.). Sediment inputs can therefore have substantial impacts on the structure and functioning of aquatic ecosystems, which may ultimately affect their provision of ecosystem services to local communities (Mitsch, W.J., et al. 2015. Ecosystem services of wetlands. International Journal of Biodiversity Science, Ecosystem Services & Management, 11, 1-4.).

Response

In the new version of the manuscript, we included a paragraph (lines 366-372) where the information regarding the effect of sediment on aquatic systems is complemented and supported with the literature recommended by the reviewer as well as other studies.

Line 378: This is an important point, because fish communities are known to be influenced by the forests within their catchment (Lo, M., et al. 2020. The influence of forests on freshwater fish in the tropics: A systematic review. BioScience, 70, 404-414.), and so the conversion of native rainforest to oil palm plantations will likely have impacts on the in-stream fish communities in this region.

Response

We have included a paragraph (lines 389-392) which better describes the influence of forest on fish communities using this as it is an interesting example of the impact of deforestation on freshwater fishes.

Lines 762-766: There is an error with the formatting of the references here. There are currently two reference #37s and two reference #38s.

Response

Thank you very much, we removed the two references that were not cited in the text.

REVIEWER 2

GENERAL COMMENTS

The expansion of oil palm plantations is a global issue. This study used Landsat 8 OLI images coupled with regular geostatistics to quantify the spatial distributions of oil palm area and the land-use changes of different landscapes (agricultural, oil palm crop, aquatic ecosystem, etc..) over 17-year plantation history (from 2001-2017). As a case study, the present work alerts the audiences to the serious problem in a biodiversity hotspot (Usumacinta River Basin), in face to the accelerated oil palm plantation processes.

We thank the reviewer for the time spent to review this manuscript and for the comments.

MAJOR COMMENTS

(1) Although the present study has used several geostatistical tools to reveal the landscape patterns in the Usumacinta watershed, the effects of oil palm crops on floodplain ecosystems remained largely descriptive, without substantial data supporting them (e.g., lines of the evidence mostly derived from previous pieces of literature). Therefore, the objectives of this study should be better framed.

Response

We very much agree with the reviewer. We have included a landscape analysis which better describes the effects of oil palm on floodplain ecosystems (Lines 236-273; 393-443; 451-454). Details about the specific changes can be seen in the following responses.

(2) To well address the above issue, I suggest further quantifying the fragmentation degree of landscape and/or landscape heterogeneity characteristics, as well as other commonly-used geographical indices before and after large-scale oil palm plantation, based on the currently obtained data points. For instance, how “environmental integrity” is impaired by oil palm plantation, since the authors stated in multiple places the plantation could affect habitat complexity, environmental integrity, and connectivity, and highlighted the importance of structurally complex ecosystems of naturally-occurring tropical forest. By doing this, the authors can provide robust evidence to support the arguments about the negative impacts of oil palm plantations.

Response 

Methods

In the new version of the manuscript, we have included in the methods another section with the steps we follow for the “landscape analysis” suggest by the reviewer (Lines 236-273). We first defined ten subsections (windows) distributed throughout the study area to evaluate the differences between the composition and configuration of the landscape before and after the establishment of oil palm crop. In the main text are described the criteria used to define these subsections. We consider fragmentation, heterogeneity and diversity landscape metrics. Finally, we performed a Standardized Principal Component Analysis (PCA) to assess changes in the subsections previous defined, considering the metrics calculated.

Results

We improved the results section including the findings related with the fragmentation indices highlighting the effect of oil palm cultivation on the landscape heterogeneity (Lines 393-443). In this new version of the manuscript we include a table (table 4) with the landscape metrics obtained as well as a figure with the information related with the PCA. Supplementary material is also included in order to support and complement our main findings related with the landscape analysis (S1 figure and S1 table documents).

Discussion

We provide information in the discussion section that better support our landscape main findings (Lines 451-456).

Observation

It is important to mention that during the analysis of the landscape in the defined subsections, we found the presence of oil palm during 2001. We had not presented this finding in the previous submitted version. Although it does not represent a significant change, since there are few hectares of oil palm (2,987 ha), table 3 is updated in the new version of the manuscript. We also modified the description in the main text that corresponds with the updated table.

(3) Some technical issues should be addressed before publication: a) lines 180-181, the cloud cover data for the images used should be provided; b) Landsat 8 satellite was only launched after 2013, how can obtain images using Landsat 8 for those in 2011-2012; c) line 182, more information about atmospheric and geometrical correction should be provided. For instance, what models have been adopted for atmospheric correction?

Response

a) We have added information about the cloud data we used to select images (lines 186-187).

b) Thank you very much for this observation. In the previous manuscript we forgot to include the other type of image that we used for the 2001 classification. In the new version of the manuscript this information is included (lines 182-184)

c) We present further information in the text about the tools and models used to perform the atmospheric and geometrical correction (lines 191-203).

(4) There is a combined result and discussion in the Result section. Please make a clear separation between your results and discussion, and be sure the discussion answers what the introduction asked.

Response

We appreciate the reviewer comment. However, after reading the results, we consider that they are written quite specifically and concrete. Only in the first section referred with the ES identified and the potential effect of oil palm crops (lines 221-282) there is a broader description of the findings that we think is more suitable for the Results section rather than the Discussion.

In addition, in the new version of the manuscript, we include information in the discussion section that better support our main findings. 

If we misunderstood the suggestion, and the reviewer did not mean this, we would be very grateful if she/he could provide further explanation.

MINOR COMMENTS

1. Line 44, oil palm production in 2010 and 2011 has been reported. What about the current conditions (e.g., in 2020 or the recent five years)?

Response

We appreciate your review and comments. We include information in the new manuscript related with the current condition of oil palm production (2020). We support the data with references.

2. Lines 48-49, “8.9 billion tons in 2050”, it means globally or in Indonesia? Please specify.

Response

Yes, we referred to the global production of oil palm. We include in the text the word “global” for more clarity.

3. Line Abstract & 21 & 110, add “a” before case-study site

Response

We appreciate your review and comments. We accept these suggestions and the changes were made to the main text accordingly.

4. Lines 114-118, the study aim should be framed more logically. For example, the vertical distribution of oil palm plantations, land-use conversion, as well as the spatial distribution of oil palm crops by freshwater ecoregion have been studied, but not been clearly stated here.

Response

We appreciate your review and comments. We accept this suggestion and the text has been modified (Lines 115-121), stating more clearly the main questions related with the aim of the study.

5. Line 128, remove the comma before [52].

6. Line 129, add a colon between “floodplains” and “a deltaic one”.

7. Lines 169-170, rephrase this sentence.

Response

We accept these suggestions and the text was modified accordingly.

8. Line 203, “Table 2” should not be ahead of “Table 1”.

Response

Thank you very much for your comment. To avoid confusion with the order of the tables in the new version of the manuscript we removed the paragraph “A brief description of the land cover categories is given in Table 2”. This hopefully allows a better understanding of the text.

9. Lines 251-252, your data points cannot support your statements “we found that the greatest environmental impact comes from pollution and habitat fragmentation due to deforestation”. Besides, a citation of this sentence means that you cited previous reports to support your viewpoints.

Response

We agree with the reviewer that our data point cannot support our statement “we found that the greatest environmental impact comes from pollution and habitat fragmentation due to deforestation”. In fact, this section is a brief description of the "potential" impacts of oil palm on riverine floodplains and is based on few studies we found, but we believe it helps introduce the reader to the issue of oil palm and its effects.

We modified the text including only an overall description of the oil palm impacts supported with the reviewed studies (lines 253-255).

10. Line 301, from Table 3, how can the increased oil palm areas be contributed to a net loss of 39% area of rainforest?

Response

Thank you very much for this observation. Indeed, this data (39% net loss) cannot be observed in table 3 so we remove it from the text in this new version of the manuscript. In fact, this data is mentioned in the section where we describe the land cover that was converted to oil palm (line 322-326). The percentage converted to palm oil during the study period (2001-2017) is calculated for each cover type coverage and this can be observed in figure 5. 

11. Lines 401-403, perhaps the authors can give a simple calculation to account for this conclusion. What is the balance between the production of bioenergy in form of carbon and the loss of biomass carbon?

Response

We provided information on this issue supported with the relevant literature (lines 414-423). If we misunderstood the suggestion, and the reviewer did not mean this, we would be very grateful if she/he could provide further explanation.

---

## [Decision Letter · Decision Letter 1]

25 Mar 2022

Tropical wetlands and land use changes: The case of oil palm in neotropical riverine floodplains.

PONE-D-21-26813R1

Dear Dr. Camacho-Valdez,

We’re pleased to inform you that your manuscript has been judged scientifically suitable for publication and will be formally accepted for publication once it meets all outstanding technical requirements.

Kind regards,

Bijeesh Kozhikkodan Veettil

Academic Editor

PLOS ONE

Additional Editor Comments (optional):

Reviewers' comments:

Reviewer's Responses to Questions

**Comments to the Author**

1. If the authors have adequately addressed your comments raised in a previous round of review and you feel that this manuscript is now acceptable for publication, you may indicate that here to bypass the “Comments to the Author” section, enter your conflict of interest statement in the “Confidential to Editor” section, and submit your "Accept" recommendation.

Reviewer #1: All comments have been addressed

Reviewer #2: All comments have been addressed

2. Is the manuscript technically sound, and do the data support the conclusions?

Reviewer #1: Yes

Reviewer #2: Yes

3. Has the statistical analysis been performed appropriately and rigorously? 

Reviewer #1: Yes

Reviewer #2: Yes

4. Have the authors made all data underlying the findings in their manuscript fully available?

Reviewer #1: Yes

Reviewer #2: Yes

5. Is the manuscript presented in an intelligible fashion and written in standard English?

Reviewer #1: Yes

Reviewer #2: Yes

6. Review Comments to the Author

Reviewer #1: Thank you for undertaking these thorough and well-documented revisions to your study. The revised manuscript is a clear improvement, and in my view makes an important contribution to the literature on the responses of wetland ecosystems to land use changes.

Reviewer #2: I am glad to recommend acceptance for the publication of the manuscript based on the detailed and satisfactory modifications made. Before reaching final decisions, I sincerely hope the authors check the manuscript again for typos or any potential grammar mistakes involved. For instance, in the text, " table 4 " should be " Table 4 "; in the caption of Table 4, " 2001-201 " should be " 2001-2017 "; for the term " 2X2 km ", there is no space between numeric and multiple sign (i.e., X), but for " 24 X 24 ", it has. Multiple places should be re-visited again to meet the high-quality requirement of the Journal.

7. PLOS authors have the option to publish the peer review history of their article (what does this mean?). If published, this will include your full peer review and any attached files.

Reviewer #1: No

Reviewer #2: No

---

## [Editor Report · Acceptance letter]

31 Mar 2022

PONE-D-21-26813R1 

Tropical wetlands and land use changes: The case of oil palm in neotropical riverine floodplains. 

Dear Dr. Camacho-Valdez:

I'm pleased to inform you that your manuscript has been deemed suitable for publication in PLOS ONE. Congratulations! Your manuscript is now with our production department. 

Kind regards, 

on behalf of

Dr. Bijeesh Kozhikkodan Veettil 

Academic Editor

PLOS ONE